# Mechanism of glycoform specificity and in vivo protection by an anti-afucosylated IgG nanobody

Aaron Gupta[1,6], Kevin S. Kao[1,6], Rachel Yamin[1], Deena A. Oren [2], Yehuda Goldgur[3], Jonathan Du[4], Pete Lollar[5], Eric J. Sundberg [4] & Jeffrey V. Ravetch [1] ✉

Immunoglobulin G (IgG) antibodies contain a complex *N*-glycan embedded in the hydrophobic pocket between its heavy chain protomers. This glycan contributes to the structural organization of the Fc domain and determines its specificity for Fcγ receptors, thereby dictating distinct cellular responses. The variable construction of this glycan structure leads to highly-related, but non-equivalent glycoproteins known as glycoforms. We previously reported synthetic nanobodies that distinguish IgG glycoforms. Here, we present the structure of one such nanobody, X0, in complex with the Fc fragment of afucosylated IgG1. Upon binding, the elongated CDR3 loop of X0 undergoes a conformational shift to access the buried *N*-glycan and acts as a 'glycan sensor', forming hydrogen bonds with the afucosylated IgG *N*-glycan that would otherwise be sterically hindered by the presence of a core fucose residue. Based on this structure, we designed X0 fusion constructs that disrupt pathogenic afucosylated IgG1-FcγRIIIa interactions and rescue mice in a model of dengue virus infection.

Glycans are ubiquitous and play essential roles throughout biology. They are present in all living cells, decorating proteins, lipids, and even nucleic acids, where they help to define molecular functions. On glycoproteins, glycans often exist as structurally similar but distinct motifs, known as glycoforms. Although related protein glycoforms have an identical amino acid backbone and may differ by only a single sugar residue, they can exhibit vastly different biological functions. In this vein, some glycoforms may be markers of specific disease states or differentially contribute to protective and pathogenic mechanisms, as has been documented extensively[1–9]. This makes precision targeting of protein glycoforms an attractive platform for clinical diagnostics and therapeutics.

Immunoglobulin G (IgG) antibodies are one such example of glycoproteins that exhibit considerable glycoform diversity. The IgG Fc domain, a homodimer, is decorated with a single complex biantennary *N*-glycan at N297 in the Cγ2 domain on each protomer, which can adopt one of thirty-six states; in combination with subclass sequence diversity, glycan heterogeneity provides hundreds of structural permutations of IgG[10]. The core heptasaccharide of this glycan consists of two proximal N-acetyl glucosamine (GlcNAc) residues and one mannose residue connected by β−1,4 linkages, two branching mannose residues (one α−1,3, one α−1,6), and two additional terminal GlcNAc residues connected by β−1,4 linkages. This core $Man_3GlcNAc_4$ motif is present on every IgG molecule regardless of subclass and is highly conserved across species. Most importantly, it is essential for binding to Fcγ receptors (FcγRs), a family of activating and inhibitory cellular receptors for IgG that initiate a wide array of downstream protective and pathogenic effector functions, including tumor and

[1]Laboratory of Molecular Genetics & Immunology, The Rockefeller University, New York, NY, USA. [2]Structural Biology Resource Center, The Rockefeller University, New York, NY, USA. [3]Structural Biology Program, Memorial Sloan Kettering Cancer Center, New York, NY, USA. [4]Department of Biochemistry, Emory University School of Medicine, Atlanta, GA, USA. [5]Department of Pediatrics, Emory University School of Medicine, Atlanta, GA, USA. [6]These authors contributed equally: Aaron Gupta, Kevin S. Kao. ✉e-mail: ravetch@rockefeller.edu

pathogen clearance, removal of infected or cancerous cells, modulation of lymphocyte responses, and the initiation of anti-inflammatory pathways that restrain the immune response[11–13].

To generate glycoform diversity, the core heptasaccharide can be extended with core fucose, bisecting GlcNAc, terminal galactose, and/or terminal sialic acid residues. Together, these secondary glycan modifications further modulate Fc-FcγR binding and contribute to the specificity of Fc-FcγR interactions, directing binding to specific FcγRs to execute critical antibody effector functions[14–16]. The presence or absence of the core fucose, attached to the most proximal GlcNAc of the N-glycan, modulates the binding affinity to FcγRIIIa/b. Notably, IgG lacking its core fucose has 10-40-fold higher affinity for the activating FcγRIIIa due to glycan-glycan interactions between ligand and receptor[17–19]. Fc fucosylation is regulated during inflammation and infection, playing a role in pathogen clearance and, when dysregulated, in pathogenicity. For example, in secondary dengue infection, afucosylated IgG1 titers – and therefore the degree of FcγRIIIa engagement – correlates with and causes disease severity[20]. In some cases of secondary dengue infection, titers of afucosylated IgG1 at the time of hospital admission predict severe disease manifestations later in the clinical course, making Fc glycan structure a valuable prognostic[21]. Similarly, enveloped viruses such as SARS-CoV-2 and CMV induce afucosylated IgG1 that is correlated with and may predict disease outcomes[22–24]. Studies have also shown this phenomenon in autoimmune diseases, such as neonatal alloimmune thrombocytopenia, where anti-platelet antibodies are significantly afucosylated compared to total IgG and are correlated with clinical severity[25]. In other cases, titers of afucosylated IgG can mediate a protective effect, as has been shown in acute viral or malaria infection[26,27]. This pathway has been exploited in therapeutic antibody development with afucosylated IgG antibodies engineered for enhanced FcγRIIIa affinity resulting in augmented cellular cytotoxicity to increase the efficacy of monoclonal antibody therapeutics[28–30].

Unlike other glycoproteins, IgG harbors two glycans which are buried in the cleft between the two heavy chains, and are thus not solvent exposed or readily accessible on the surface of the molecule to conventional probes. We previously described an approach to overcome this challenge and successfully identified synthetic nanobodies that can distinguish fucosylated and afucosylated IgG glycoforms[31]. These nanobodies achieved striking specificity, with no observed cross-reactivity to off-target Fc glycoforms, glycoproteins, or free glycans, constituting a novel class of biologic agents. However, the structural basis for their glycoform specificity remains unknown. In the present study, we present the crystal structures of the afucosylated IgG-specific nanobody X0 in isolation, as well as in complex with afucosylated IgG1 Fc. We describe a unique mode of recognition of afucosylated glycoforms which relies on protein-protein contacts formed by all three X0 CDRs and reveals the long and flexible CDR3 as a sensor for the buried Fc glycan. In addition, we exploited this structural information to develop X0 as a therapeutic to disrupt pathogenic afucosylated IgG1 Fc-FcγRIIIa interactions in a mouse model of antibody-dependent enhancement of dengue virus infection. These structural and functional studies characterize the mechanism and utility of this class of nanobodies and provide insights into the design of additional glycoform-specific reagents that may have broad clinical impact.

## Results

### Crystal structure of the afucosylated IgG-specific nanobody X0
Several structures of nanobodies derived from the synthetic camelid nanobody library used in this study have been reported in the literature[32,33]. However, glycoform-specific nanobodies deviate in CDR sequence from these published models, likely resulting in significantly different loop architecture and requiring that we solve the structure of our clones in isolation. We crystallized an intermediate affinity

(~142 nM) afucosylated IgG-specific clone, X0, and determined its structure by molecular replacement, using its predicted structure from AlphaFold 2 as a search model[34,35], to a resolution of 1.8 Å (Supplementary Fig. 1A and Table 1). The loop architecture of the crystal structure for CDR1 (residues 26–34) and CDR2 (residues 46–58) aligned very closely with its predicted structure, however, electron density for CDR3 (residues 91–108) deviated considerably and packed closer to the globular Ig fold of the nanobody than predicted. Consistent with the previously mentioned characteristics of camelid nanobodies, the CDR3 loop protruded from the main body of the molecule, indicating its inherent flexibility and potential to reach recessed epitopes, such as the N-glycans buried in the cleft of IgG molecule[36].

### Architecture of the X0-afucosylated IgG1 Fc complex
To define the molecular interactions between afucosylated IgG Fc-specific nanobodies and IgG1, we determined two crystal structures of X0 in complex with afucosylated IgG1 Fc (Fig. 1a). Co-crystallization of IgG1 Fc is typically difficult due to the propensity of the Fc fragment to crystallize independently of its binding partners. To overcome this potential hurdle, we generated E382X mutants (E382A, E382R, and E382S) that have recently been reported to reduce salt-bridge interactions between E382 and R255 that are commonly seen in the $P2_12_12_1$ space group (such as PDB: 3AVE), more easily allowing for non-canonical crystal packing arrangements[37]. The C2 structure was determined by molecular replacement and multiple rounds of refinement using unliganded Fc (PDB 3AVE) and our X0 structure described above as search models. The $P6_1$ structure was determined by molecular replacement using the C2 structure as a search model. In both complexes, electron density resolved all amino acids of X0, residues 237–444 of IgG1 Fc, and the core N-linked heptasaccharide at N297 on both Fc chains. Additionally, a lack of electron density at the position of the core fucose corroborated the afucosylation status of our IgG Fc molecules. The final structures were refined to 2.6 Å for C2 and 2.7 Å for $P6_1$ (Table 1), and were nearly identical structurally, with an RMSD of 1.5 Å. The only differences observed between the two models are due to crystal packing, resulting in a domain shift in IgG1 around the hinge region (residues 342–343). To verify this, we independently superimposed the N- and C-terminal domains of the two models, revealing an RMSD between Cα atoms of 0.374 Å (N-term) and 0.333 Å (C-term). The contact area between X0 and IgG1 Fc is unaffected. For all figures presented, the $P6_1$ structure was used.

In both crystal structures, the X0-IgG Fc complex consists of two X0 nanobodies bound to one Fc homodimer, reflecting ability of X0 to recognize both open surfaces of the symmetrical Fc domain. Because the Fc fragment was used for these structural studies, it is possible that the presence of Fab domains in a full-length afucosylated antibody would prevent secondary X0 binding. To address this, we confirmed the stoichiometry of this interaction by sedimentation velocity analytical ultracentrifugation and obtained a value of 73.8 kDa, in reasonable agreement with the 2:1 X0:IgG Fc sequence molecular weight, adjusted for the Fc glycan, of 79.6 kDa (Supplementary Fig. 2). This mode of IgG recognition by the X0 nanobody contrasts with that of FcγRs, which asymmetrically intercalate within the groove of the Fc homodimer, precluding binding of a second FcγR molecule and resulting in a complex consisting of one FcγR bound to one Fc homodimer.

### All CDR loops of X0 interact with the protein backbone of IgG1 Fc
Superposition of the apo and bound structures of X0 revealed minimal changes in the CDR1 and CDR2 loop position upon binding to IgG Fc. However, consistent with an induced-fit model, the CDR3 loop undergoes a large conformational shift, curling downwards and wrapping around the IgG C'E loop to gain better access to the buried N-

glycan at N297. This is demonstrated by several CDR3 residues that in some cases move >9 Å upon binding (Fig. 1c). All three CDR loops of X0 contact the Fc protein backbone (Fig. 2a). Notably, all hydrogen bonds between X0 and the Fc fragment occur within the C′E loop of the Fc, which harbors the *N*-glycan and is a region critical for FcγR binding. Within the CDR1 loop, the sidechain of Y31 forms a hydrogen bond with $O\epsilon^2$ of E269. T33 forms hydrogen bonds with the sidechains of Y296 and S298 as well as the main chain nitrogen of S298. Y37 forms a hydrogen bond with Y296 of the Fc (Fig. 2b). In the CDR2 loop, the main chain nitrogen of W53 forms a hydrogen bond with the side chains of E294 and S298 (Fig. 2C). The main chain nitrogen of G54 hydrogen bonds with $O\epsilon^2$ of E294. Finally, in the CDR3 loop, $O\delta^2$ of D98 hydrogen bonds with Y296, while the main chain oxygen of G100 forms a hydrogen bond with $N\delta^2$ of N297 (Fig. 2d).

X0 was generated through affinity maturation of a previously published low-affinity parental clone, C11[31]. We previously demonstrated that this clone is high affinity and specific for afucosylated IgG1 (G2: $K_D$ 142 nM, G2F: $K_D$ 3.7 μM). During this process, some residues were conserved across all clones and we suspected these were important for high affinity X0-IgG1 Fc binding (Supplementary Fig. 3). The importance of some of these residues (F47, T52, and Y58 in CDR2; Y106 in CDR3) as well as select aforementioned X0 binding residues was experimentally validated by generating single alanine mutants at the indicated positions. Similarly, we generated afucosylated alanine

mutants of the putative binding residues of IgG1 Fc, as well as other residues known to be important for FcγR binding: BC loop (H268, E269), C′E loop (E294, Y296), and FG loop (L328, P329, I332). The relative positions of these putative interactions are highlighted (Fig. 2a–d), and the impact of all mutants are shown as $\log_2$ fold change in $K_D$ relative to wildtype X0-Fc binding, as measured by surface plasmon resonance (SPR) (Fig. 2e). All X0 mutants reduced binding to afucosylated IgG by 4- to 30-fold and binding to fucosylated IgG1 Fc was undetectable by SPR. C′E loop mutants completely abrogated binding, demonstrating that these residues are individually critical for X0's recognition of the Fc. The importance of interactions with Y296 may explain, in part, X0's unfavorable binding to afucosylated IgG2-4. For example, both IgG2 and IgG4 contain Phe at position 296 (Supplementary Fig. 3B). This substitution likely disrupts hydrogen bonding with T33, Y37, and D98 of X0. In contrast, both BC loop and FG loop contacts were, in isolation, dispensable for binding.

### X0 CDR3 functions as the sensor for afucosylated IgG Fc glycoforms

Two residues in the X0 CDR3 loop, G100 and T101, interact with the IgG glycan (Fig. 3a and Supplementary Fig. 4). Both of these residues were conserved across all afucosylated-specific nanobody clones, regardless of affinity. G100 and T101 are positioned at the apex of the CDR3 loop in close proximity to the *N*-glycan. The main chain N atom

## Table 1 | Data collection and refinement statistics

| | Nanobody | Complex I | Complex II |
|---|---|---|---|
| Data collection | | | |
| Beamline | NSLS-II FMX | NSLS-II AMX | NSLS-II AMX |
| Space group | $I2_12_12_1$ | $C2$ | $P6_1$ |
| Cell dimensions | | | |
| $a, b, c$ (Å) | 59.3, 95.1, 110.4 | 125.6, 92.2, 76.5 | 170.6, 170.6, 126.2 |
| $\alpha, \beta, \gamma$ (°) | 90, 90, 90 | 90, 117.0, 90 | 90, 90, 120 |
| Resolution (Å) | 50–1.8 (1.84-1.8) | 50–2.6 (2.64–2.6) | 50–2.7 (2.75–2.70) |
| Wavelength (Å) | 0.97934 | 0.92010 | 0.92010 |
| $R_{pim}$ | 0.045 (0.339) | 0.050 (0.574) | 0.035 (0.522) |
| CC(1/2) | 0.991 (0.708) | 0.996 (0.504) | 1.000 (0.604) |
| <I > / < σI > | 29.9 (1.8) | 23.6 (1.4) | 31.1 (1.5) |
| Completeness (%) | 99.2 (95.8) | 90.7 (90.4) | 98.2 (98.9) |
| Redundancy | 9.5 (6.7) | 2.5 (2.5) | 4.6 (4.8) |
| Unique reflections | 27855 | 21352 | 56382 |
| Phasing | | | |
| Search model | Alphafold | 3AVE+nanobody | complex I |
| Refinement | | | |
| $R_{work}/R_{free}$ | 0.192//0.222 (0.289 /0.290) | 0.208/0.279 (0.355/0.450) | 0.220/0.255 (0.363/0.352) |
| $B$-factors (Å$^2$) Average/Wilson | 31.2/23.4 | 78.1/60.2 | 94.0/71.1 |
| RMS deviations | | | |
| Bond lengths (Å) | 0.007 | 0.015 | 0.012 |
| Bond angles (°) | 0.959 | 1.95 | 2.06 |
| Ramachandran plot | | | |
| % favored | 98.3 | 96.5 | 97.5 |
| % allowed | 1.7 | 3.5 | 2.4 |
| % outliers | 0 | 0 | 0.1 |
| Model contents | | | |
| Protomers/ASU | 2 | 2 | 4 |
| Protein residues | 240 | 653 | 1308 |
| Water | 175 | 10 | 14 |
| PDB ID | 8F8V | 8F8X | 8F8W |

Values in parentheses refer to the highest resolution shell. R$_{free}$ set consists of 5% of data chosen randomly against which structures was not refined.

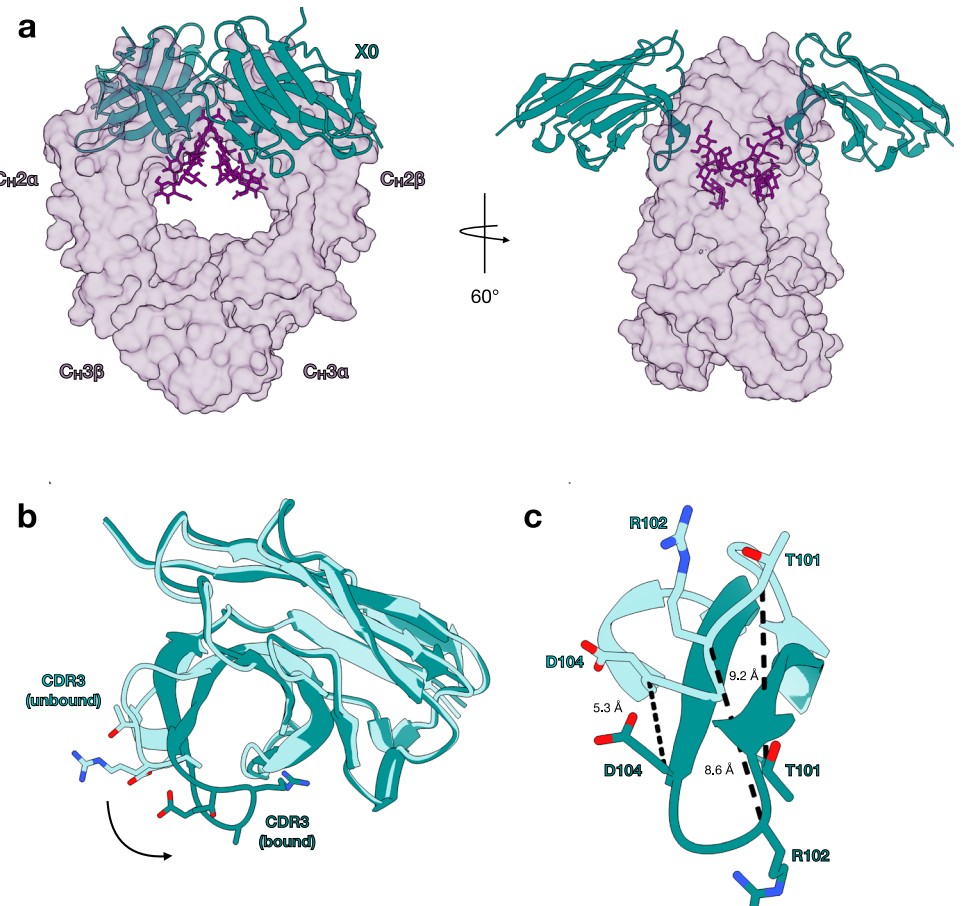

**Fig. 1 | Overall crystal structure of the X0-afucosylated IgG1 complex. a** Side-view (left) and top-view (right) of the complex with transparent surfaces of afucosylated IgG1 Fc (purple) and superimposed cartoons of X0 (teal) shown. *N*-glycans are shown as sticks in dark purple. **b** Unbound X0 (aquamarine) and X0 in complex (teal) superimposed, with sticks of apical CDR3 loop sidechains shown. **c** Overlay of CDR3 loops only with coloring the same as in **b**. Distances between alpha carbons of indicated sidechains shown, demonstrating an induced fit.

of G100 forms a hydrogen bond with the O7 atom of the most proximal GlcNAc (+1). In addition, the O$\gamma^1$ atom of T101 forms a hydrogen bond with the O6 atom of the same GlcNAc residue (Fig. 3b). Importantly, this is the GlcNAc residue that is variably bonded to core fucose across all IgG glycoforms. Superimposition of the X0-IgG1 Fc complex with a previously determined fucosylated IgG1 Fc structure (i.e. PDB 3AVE) revealed that a steric clash between the core fucose and T101 would result (Fig. 3c). Accordingly, the absence of a core fucose provides a key recognition pocket for X0 which is obstructed in fucosylated IgG antibodies, suggesting that this is the mechanism governing X0 nanobody glycoform specificity (Fig. 3b, c). Consistent with its conservation during affinity maturation of C11, site-saturation mutagenesis at position 101 showed reduced binding across all mutants (Fig. 3D). Mutants with residues of similar size (Ile, Ser) or ability to form hydrogen bonds (Gln, Glu) were generally more favorable than others, but nevertheless there was a clear preference for Thr.

### X0 CDR3 recapitulates the unique carbohydrate-carbohydrate interactions between FcγRIIIa and afucosylated IgG1

Numerous structures of IgG Fc-FcγR complexes have demonstrated asymmetric binding by FcγRs, which result in a more open conformation of the Fc, with one of the immunoglobulin domains of FcγR nestled into the groove at the IgG hinge-Cγ2 interface[38–41]. Superimposition of the X0-Fc complex and an afucosylated IgG1 Fc-FcγRIIIa complex (PDB: 3SGK) revealed overlapping epitopes between X0 and FcγRIIIa, agreeing with previously published inhibition studies[31]

(Fig. 4a). FcγRIIIa's preference for afucosylated IgG1 Fc is a result of interactions between its oligomannose glycan at N162 and the complex *N*-glycan of IgG Fc[17,42]. Specifically, hydrogen bonds are formed between the first two core GlcNAc residues of the receptor and the most proximal GlcNAc of the IgG Fc. Specifically, O3 of GlcNAc1 (FcγRIIIa) binds O7 of GlcNAc (+1) (IgG1 Fc) and O6 of GlcNAc2 (FcγRIIIa) binds O6 of GlcNAc (+1) (IgG1 Fc). These glycan-glycan interactions are weakened and/or non-existent when binding fucosylated species. In our X0-Fc structure, the X0 CDR3 loop occupies a nearly identical space as the FcγRIIIa-N162 glycan, recapitulating these exact interactions (Fig. 4b, c).

### X0-Fc fusions prevent antibody-dependent enhancement of dengue infection

Given previous data demonstrating the capacity for nanobody-mediated blockade of Fc-FcγR interactions[31], we explored whether clone X0, with intermediate affinity and high specificity, could be used in vivo as a therapeutic. One phenomenon that implicates Fc-FcγR interactions in disease pathogenesis is antibody-dependent enhancement (ADE) of dengue virus (DENV) infection. In humans, dengue infection often manifests as mild or inapparent disease. However, 10% of patients progress to more severe forms of disease, known as dengue hemorrhagic fever (DHF) and dengue shock syndrome (DSS). In these cases of ADE, rather than contributing to antiviral immunity, pre-existing antibodies generated during primary infection facilitate viral entry and subsequent infection of host cells, leading to both increased

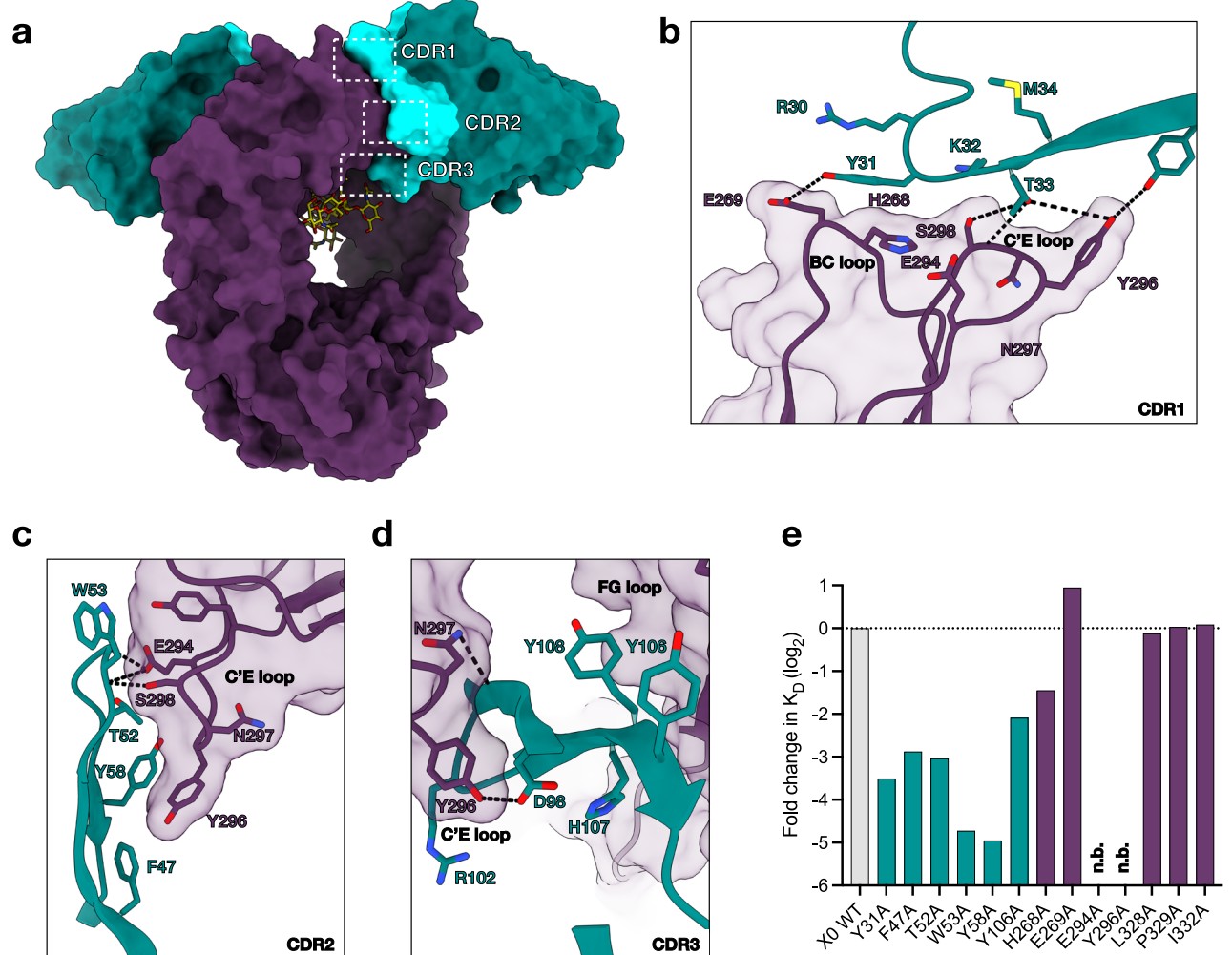

**Fig. 2 | Protein-protein contacts at the nanobody-IgG binding interface.**
**a** Overall structure of the complex showing surfaces of X0 framework regions
(teal), X0 CDR loops (aquamarine), afucosylated IgG1 Fc (dark purple), and stick
representation of the *N*-glycans in yellow. CDR loops are highlighted by dashed
white boxes. **b–d** Contact residues of CDR1, CDR2, and CDR3 with the afucosylated
IgG1 Fc protein backbone. Hydrogen bonds shown as dashed black lines and bond
lengths indicated. **e** Log$_2$ fold change in $K_D$ of X0 and Fc alanine mutants of putative
contact sites. Horizontal dashed line represents the $K_D$ of the wildtype X0-
afucosylated IgG1 Fc interaction. N.b. indicates no binding. Source data are pro-
vided as a Source Data file.

infectivity and virulence. This occurs when anti-DENV antibodies, at
sub-neutralizing titers, complex with the DENV virion and attach to the
surface of FcγR-expressing leukocytes, utilizing the phagocytic FcγR
pathway for entry[43–45]. In agreement with these in vitro observations, a
pathogenic role for antibodies in dengue has been demonstrated
in vivo in mouse and non-human primate disease models using poly-
clonal IgG isolated from symptomatic dengue patients or monoclonal
anti-dengue IgG[46–51].

One recent study from our laboratory generated a novel mouse
model of ADE that is both permissive to DENV infection and reca-
pitulates human Fc-FcγR interactions in vivo[52]. This study identified
afucosylated IgG1-FcγRIIIa interactions as the chief determinant of
ADE severity. While pre-treatment of mice with Fc null anti-DENV
monoclonal antibodies (clone C10) protects against dengue infec-
tion, afucosylated IgG1 enhances disease, evident from severe
weight loss, thrombocytopenia, and eventual death (Fig. 5a). We
sought to disrupt this glycoform-specific Fc-FcγRIIIa interaction and
prevent ADE in these mice. Pre-treatment with X0-Fc rescued mice
from thrombocytopenia, severe weight loss, and death compared to
isotype controls (Fig. 5b–d), demonstrating that X0 is a potential
therapeutic that can selectively target afucosylated IgG glycoforms
and prevent ADE.

## Discussion

Glycoproteins and glycopeptide antigens are attractive targets for
diagnostics and therapeutics. However, approaches to target these
molecules have fallen short, as antibodies against carbohydrate anti-
gens, as well as natural carbohydrate binding proteins, like lectins,
suffer from low-binding affinities and/or poor specificity[53–55]. Pre-
viously, we generated glycoform-specific nanobodies from a synthetic
yeast display library. While we attributed their specificity to the long
and flexible CDR3 domain present in camelid heavy-chain variable
domains (VHHs), the structural basis of glycoform recognition was not
readily evident.

In the current study, we describe the mode of recognition of an
afucosylated IgG1-specific nanobody, X0. This specificity relies on two
key structural properties: (i) protein-protein interactions formed by all
three CDR loops of the nanobody with the protein backbone of the Fc,
and (ii) a 'glycan sensor' facilitated by the highly flexible CDR3. While the
protein-protein interactions by the various CDR loops are necessary for
preserving affinity of these molecules, the ability of these nanobodies to
discriminate between afucosylated and fucosylated Fc glycoforms is
driven by the CDR3 loop. This recognition is mediated by two residues
at the apex of the CDR3 loop, G100 and T101. In particular, T101 makes a
critical hydrogen bond with GlcNAc(+1) on the Fc glycan, which would

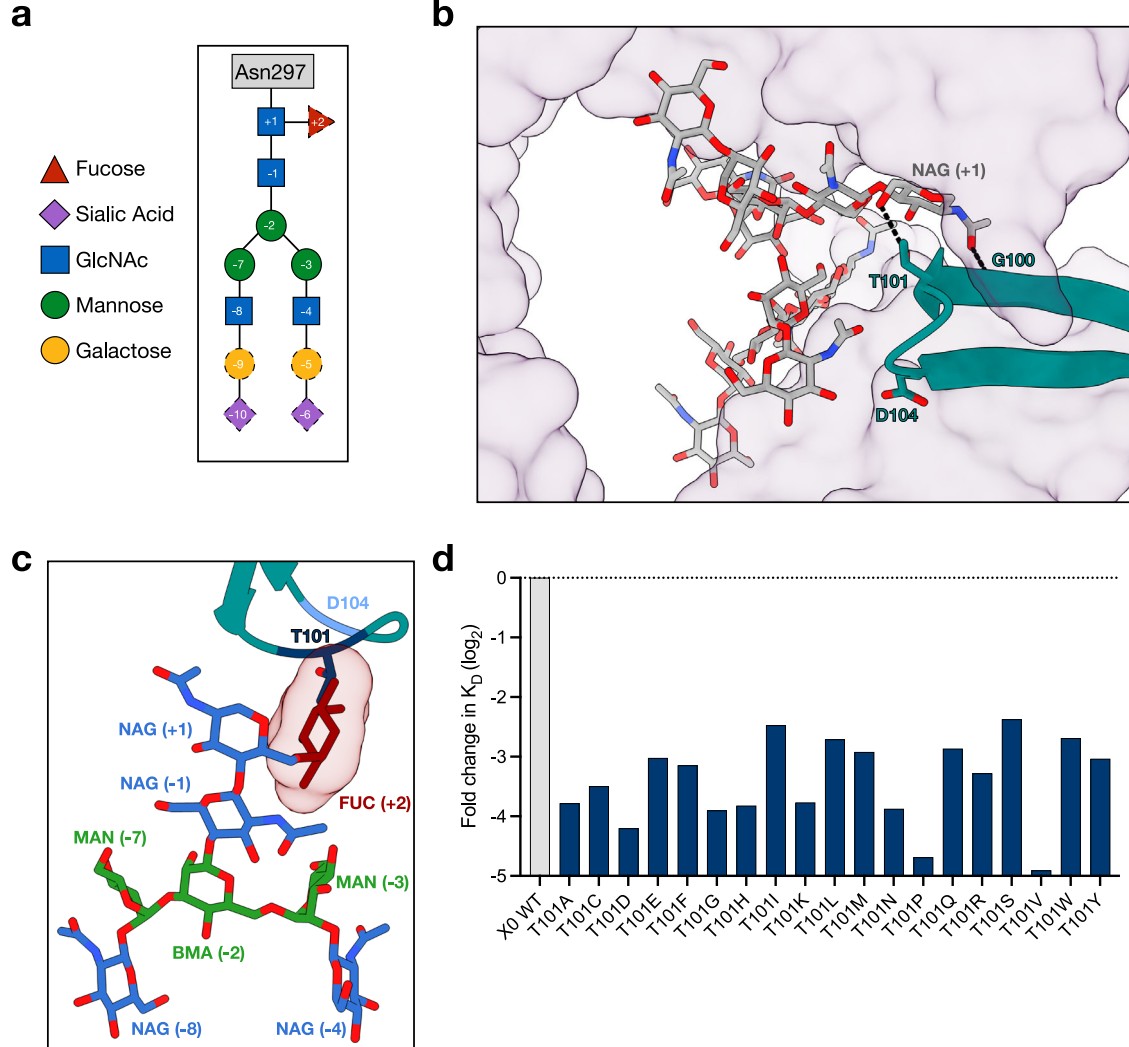

**Fig. 3 | Protein-glycan contacts at the nanobody-IgG binding interface.**
**a** Schematic of the highly conserved *N*-glycan at IgG Fc N297. Residues colored and numbered according to SNFG standards. Residues with solid outline are part of the core $Man_3GlcNAc_4$ motif, while those with dashed outlines are variable. **b** X0 CDR3 loop contacts the Fc glycan. Hydrogen bonds are shown by dashed black lines. IgG1 as transparent surface (purple), X0 as cartoon (teal), and glycan as sticks with heteroatoms shown (yellow). Relevant X0 residues highlighted in shades of blue. **c** The X0 CDR3-Fc glycan interface with the core fucose (red sticks with transparent surface) modeled in from 3AVE to reveal potential clashes. **d** $Log_2$ fold change in $K_D$ of X0 T101 mutants. Horizontal dashed line represents the $K_D$ of the wildtype X0-Fc interaction. Source data are provided as a Source Data file.

otherwise be sterically occluded by the presence of a core fucose. Given the importance of the CDR3 in glycoform discrimination, it is possible that nanobodies targeting other glycoproteins with solvent unexposed or recessed epitopes could be isolated, as similar findings have been demonstrated with the unique ultralong CDR3 architecture of bovine monoclonal antibodies and HIV human bnAbs[56–58].

In contrast to the canonical type I receptors for IgG Fc, FcγRs, for which one FcγR binds asymmetrically to one Fc homodimer, our X0 nanobody binds symmetrically with two X0 molecules binding equivalent sites on each of the two Fc protomers. Furthermore, X0 nanobody binding to IgG Fc precludes FcγR binding in vivo, allowing for disruption of pathogenic Fc-FcγR interactions, such as those observed in ADE of dengue virus infection and potentially those driving autoimmune disease[25,59,60]. While we demonstrate direct blockade as a feasible approach for eliminating pathogenic Fc-FcγR interactions, it may also be possible to use the structure of the X0-IgG1 Fc complex to rationally design nanobody-glycosidase or protease fusions, to selectively deplete specific IgG glycoforms while preserving protective host antibodies[61].

To date, there have been few examples of antibodies with glycoform-level specificity[62]. Based on our structural studies, we suspect that nanobodies may have a key advantage in recognition of buried glycan structures due to their elongated CDR3 loop. In the case of IgG, the X0 CDR3 loop can insert into the hydrophobic cleft between the two Cγ2 domains of the Fc and make hybrid protein-glycan contacts which drive specificity. Future attempts to design glycoform-specific nanobodies may benefit from further elongation and diversification of CDR3 loops to access obscured glycans. In summary, this work describes a structural basis for IgG glycoform recognition and may provide key insights into the design of future molecules.

## Methods

All in vivo experiments were performed in compliance with federal laws and institutional guidelines and have been approved by the Rockefeller University Institutional Animal Care and Use Committee. Mice were bred and maintained at the Comparative Bioscience Center at the Rockefeller University.

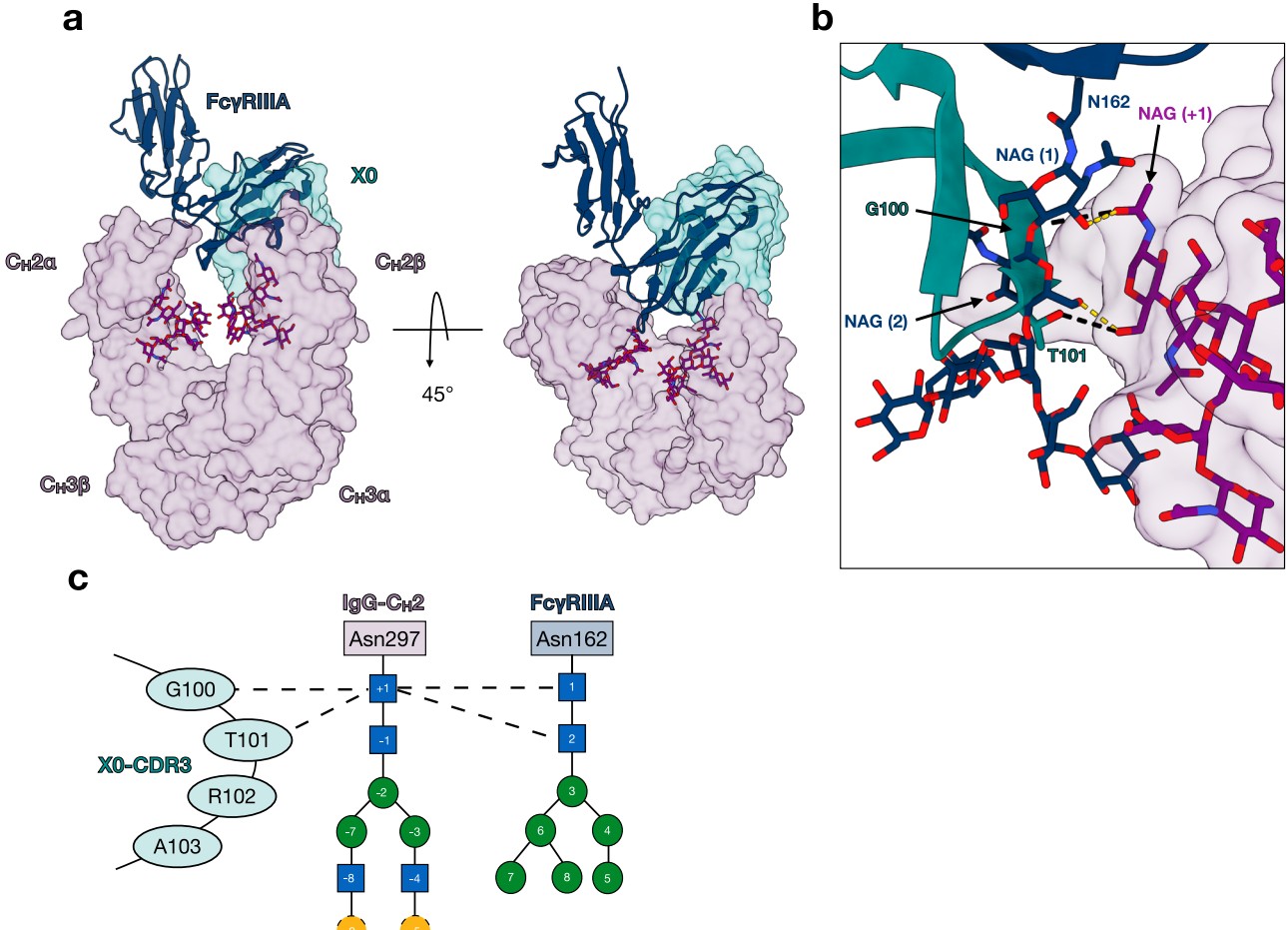

**Fig. 4 | X0 CDR3 recapitulates the unique carbohydrate-carbohydrate interactions between FcγRIIIA and afucosylated IgG1. a** Side-view (left) and top-view (right) of the superimposed complexes with transparent surfaces of IgG1 Fc (purple) and X0 (teal) shown as well as a cartoon of FcγRIIIa (navy). *N*-glycans are shown as sticks in purple with heteroatoms colored. **b** Close view of X0-CDR3 (teal), FcγRIIIa N162 *N*-glycan (navy sticks), and afucosylated IgG1 Fc *N*-glycan (purple surface). X0's G100 and T101 recapitulate the hydrogen bonds formed by the two proximal GlcNAcs of the FcγRIIIa N162 glycan. **c** Contact map describing X0 CDR3 and FcγRIIIa-glycan interactions with the Fc glycan. Dashed lines indicate hydrogen bonds.

## Expression and purification of IgG, IgG Fc, and X0-Fc

Recombinant antibodies were generated using the Expi293 or Expi293 FUT8$^{-/-}$ system (ThermoFisher) using previously described protocols[27]. An equal ratio of heavy and light chain plasmids (or only nanobody-Fc) was complexed with Expifectamine in OptiMEM and transfected into Expi293 cells in culture at $3 \times 10^6$ cells/ml. Enhancer 1 and 2 were added 20 h after transfection. Recombinant IgG was harvested and purified after 6 days with protein G sepharose beads (GE Healthcare), dialyzed in PBS, filter-sterilized (0.22 μm), concentrated with 100 kDa MWCO spin concentrator (Millipore), purified with Superdex 200 Increase 10/300 GL (GE Healthcare), and finally assessed by SDS–PAGE followed by SafeBlue staining (ThermoFisher). All antibody preparations had endotoxin levels were <0.05 EU mg$^{-1}$, as measured by the limulus amebocyte lysate assay.

## Expression and purification of nanobody X0

Nanobodies were expressed and purified as reported previously[31–33,63]. Briefly, bacteria were grown in terrific broth at 37 °C overnight. The following day, a 1:100 culture was grown until an OD of 0.7–0.9 and then induced with 1 mM IPTG. After 20–24 h of shaking at 25 °C, *E. coli* were pelleted and resuspended in SET buffer (200 mM Tris, pH 8.0, 500 mM sucrose, 0.5 mM EDTA, 1X cOmplete protease inhibitor (Sigma)) for 30 min at room temperature. Following equilibration, bacteria were osmotically lysed with the addition of 2x volume of deionized water rocked for 45 min. Prior to centrifugation at 17,000×*g*

for 20 min, NaCl was added to 150 mM, MgCl$_2$ to 2 mM, and imidazole to 20 mM. The periplasmic fraction was filtered with a 0.22 μm filter and incubated with 4 mL 50% Ni-NTA resin equilibrated in wash buffer (20 mM HEPES, pH 7.5, 150 mM NaCl, 40 mM imidazole) (Qiagen) per liter of initial bacterial culture. Supernatant and resin were incubated, rocking for an hour, and then pelleted at 50×*g* for 1 min. Ni-NTA resin was washed with 10 volumes of wash buffer before elution (20 mM HEPES, pH 7.5, 150 mM NaCl, 250 mM imidazole). Eluted protein was concentrated with 3 kDa MWCO filters (Amicon) before size-exclusion chromatography (GE Healthcare). Proteins were stable at 4 °C.

## Preparation and crystallization of X0-IgG1$^{E382S/A/R}$ Fc complexes

For the apo-X0 structure, X0 was purified from *E. coli* as described above. X0 was concentrated to 26 mg/mL and crystallized at room temp in the presence of MIDAS*plus* HT-96 (Molecular Dimensions) using the Art Robbins Phoenix crystallization dispenser, in sitting-drop format, 300 nL at 1:1 ratio. The precipitant solution contained 20% v/v PPGBA 400 + 15% v/v 1-Propanol. Crystals were submerged in 40% ethylene glycol in the precipitant solution prior to flash cooling in liquid nitrogen.

For the X0-Fc complex structure purified X0 and the specified IgG1 Fc variant were mixed at a 3:1 ratio at room temperature for 30 min. The complex was isolated through size exclusion chromatography, concentrated to 35 mg/mL crystallized in the same manner as the apo-X0, using SG1 HT-96 (*P6* complex, Molecular Dimensions) or

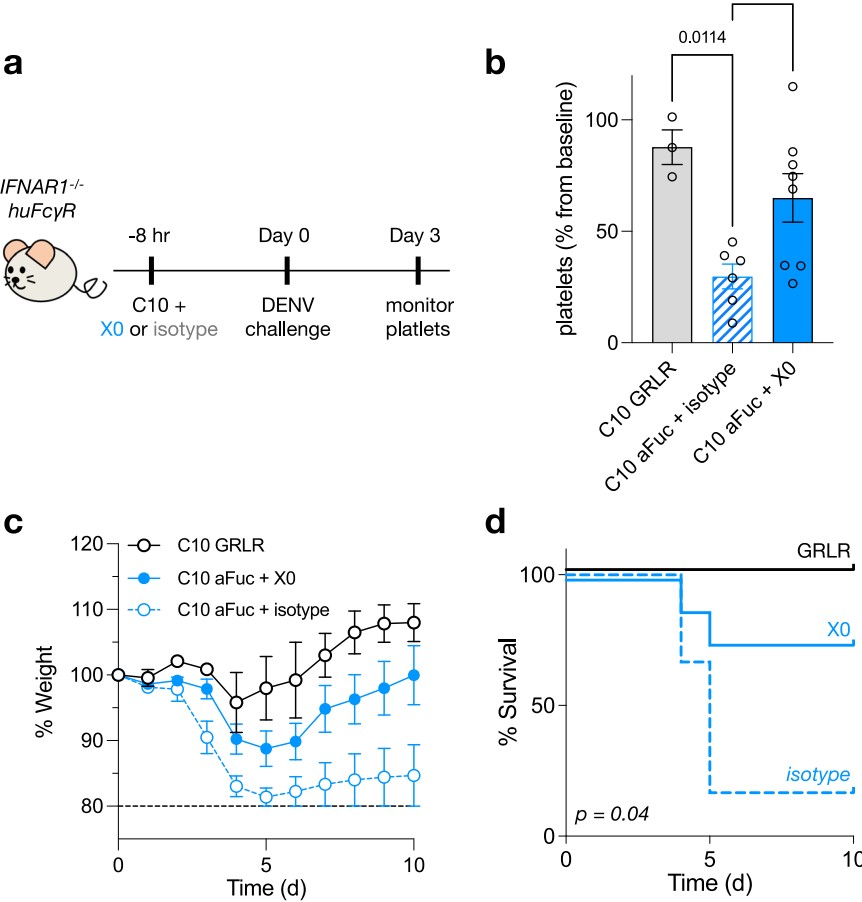

**Fig. 5 | X0-Fc fusions block afucosylated Fc-FcγRIIIa interactions to reverse antibody-dependent enhancement of dengue infection. a** 4–6-week-old mice were administered 20 μg anti-DENV C10 along with 80 μg X0-Fc or isotype i.v., followed 8 h later by i.v. dengue infection. Platelets were counted on days 0 and 3 and mice were monitored for weight loss and survival until day 10. **b** Platelet levels as a percentage of baseline. $P$ values calculated by one-way ANOVA with Bonferroni's correction for multiple comparisons. **c** Weight as a percentage of baseline at day 0. Mice were killed and excluded from further analysis if <80% of baseline. Data displayed as mean ± SEM ($n = 3$ mice for GRLR, $n = 6$ for C10 aFuc + isotype, and $n = 8$ for C10 aFuc + X0) in **b** and **c**. **d** Survival curves of mice treated in A. $P$ value comparing X0 and isotype computed by log-rank (Mantel-Cox) test. Source data are provided as a Source Data file.

LMB HT-96 (*C*2 complex, Molecular Dimensions). The precipitant solution for *C*2 contained, 20% w/v PEG 6000, 0.1 M Sodium citrate 4.0, 0.2 M Lithium chloride. No additional cryopreservation was necessary prior to flash cooled. The precipitant solution for *P*6 contained, 0.1 M Sodium HEPES 7.5 20 % w/v PEG 8000. Crystals were submerged in 12.5% glycerol in the precipitant solution prior to flash cooling in liquid nitrogen.

### Surface plasmon resonance
Surface plasmon resonance (SPR) was performed on a Biacore T200 machine (Cytiva Life Sciences). In some experiments, purified IgG glycoforms diluted in HBS-EP+ were immobilized on the surface of a Protein A or Protein G CM5 sensor chip at 1000 RU (~50 nM). Purified nanobodies were flowed over IgG-bound sensor chips at the indicated concentrations at 30 uL/min for 60 s, followed by 600 s of dissociation. Sensor chips were regenerated with 10 mM Glycine-HCl pH 1.5.

All kinetic constants were calculated using GraphPad Prism v9. For nanobody monomer binding, sensorgrams were fit using a 1:1 Langmuir binding model and kinetic constants reported. For tetramer binding, the association phase was fit separately using an association kinetics model simultaneously fitting the association rate constant for each concentration. The dissociation phase was fit to a biexponential decay model with two dissociation rate constants (one fast, one slow) shared between each concentration.

### In vivo dengue antibody-dependent enhancement
In vivo experiments were approved by The Rockefeller University Institutional Animal Care and Use Committee in compliance with federal laws and institutional guidelines. Mice were maintained at the Comparative Bioscience Center at the Rockefeller University at a controlled ambient temperature (20–25 °C) and humidity (30–70%) environment with 12-h dark:light cycle. Ifnar1−/− FcγR-humanized mice (males and females; 4–5 weeks old) were anesthetized with 3% isoflurane and given the monoclonal anti-DENV mAb (clone C10) in addition to X0-Fc or an isotype control 8 h prior to infection. For infection with DENV (New Guinea C strain) mice were given $3.5 \times 10^8$ GE, i.v. Following infection, mice were weighed daily and mortality was determined once bodyweight hit an 80% threshold, as determined by the Rockefeller Institutional Animal Care and Use Committee. Platelet counts were obtained on day 0 and day 3 and measured using an automated hematologic analyzer (Heska HT5).

### Data collection and structure refinement
X-ray diffraction data were collected from a single crystal at NSLS-II (Brookhaven National Laboratory) on beamline FMX (X0 alone) to 1.8 Å and AMX to 2.6 (*C*2) and 2.7 (*P*6₁) resolution, respectively. The data were integrated and scaled with the program HKL2000. Initial phase estimates and electron-density maps were obtained by molecular replacement with Phaser in Phenixusing an Alphafold model for obtaining the X0 structure and PDB entry 3AVE for the Fc along with

the X0 structure to solve the complex in C2. The C2 structure was used to solve the P6 dataset. Iterative model building and structural refinement was manually performed using COOT and Phenix. The quality of the final model was good as noted in Table 1. All molecular graphics were prepared with Chimera. Atomic coordinates and experimental structure factors have been deposited in the PDB under accession codes 8F8V, 8F8X, and 8F8W for X0, C2, and P6 structures.

## Analytical ultracentrifugation

Sedimentation velocity analytical ultracentrifugation was conducted using an An-50 Ti analytical rotor at 50,000 rpm (182,000×$g$) at a nominal temperature of 20 °C in a Beckman Coulter XLI analytical ultracentrifuge using standard procedures[64]. Afucosylated-Fc and X0 nanobody were run individually at nominal concentrations of 3 µM and 9 µM, respectively, and together at the same concentrations. The partial specific volumes of afucosylated-Fc and X0 nanobody were calculated based on amino acid composition using SEDFIT version 16.36 (https://spsrch.cit.nih.gov/). The partial specific volume of afucosylated-Fc was adjusted using estimated fractional carbohydrate composition of 0.056 and 0.622 mL/g as the glycan partial specific volume[65]. Samples (0.4 mL) were loaded into 12 mm pathlength Epon double sector cells equipped with sapphire windows with matched buffer (137 mM NaCl, 8.06 mM sodium phosphate, 1.94 mM potassium phosphate, 2.7 mM KCl, pH 7.4) in the reference sector. The buffer density and viscosity at 20 °C were measured using an Anton Parr DM4500 densitometer and Lovis 2000M viscometer. Absorbance scans at 280 nm were initiated after reaching the target rotor speed and collected at 4.7 min intervals. Data were corrected for scan time errors using REDATE version 1.01[66]. Data were analyzed using the continuous c(s) distribution model in SEDFIT and a sedimentation coefficient interval of 0 to 10S at 0.1S intervals[67]. Data were fitted using sequential simplex and Marquardt-Levenberg algorithms and maximum entropy regularization with a confidence interval of 0.68. The fitted parameters were c(s), the frictional ratio ($f/f_o$) and the meniscus position. Sedimentation coefficients were adjusted to the standard condition of 20 °C in solvent water. Molecular weights were calculated using the Svedberg equation as described[67].

$$M = \frac{sRT}{D(1 - \bar{v}\rho)} \quad (1)$$

where $s$ is the sedimentation coefficient, $R$ is the gas constant, $T$ is the absolute temperature, $\bar{v}$ is the partial specific volume, and $D$ is the diffusion coefficient, which was obtained from the frictional ratio using

$$D = \frac{\sqrt{2}}{18\pi} k_B T s^{-1/2} [\eta(f/f_o)]^{-3/2} \left(\frac{\bar{v}}{1 - \bar{v}\rho}\right)^{-1/2} \quad (2)$$

where $\eta$ is the solvent viscosity.

## Reporting summary

Further information on research design is available in the Nature Portfolio Reporting Summary linked to this article.

# Data availability

The structural data that supports these findings is available in the Protein Data Bank under accession numbers 8F8V, 8F8W, 8F8X, 3AVE, and 3SGK. Source data are provided with this paper.

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

## Acknowledgements

Research reported in this publication was supported by the Bill and Melinda Gates Foundation (INV-034057 to J.V.R.) and in part by the National Institute of Allergy and Infectious Diseases Grant U19AI111825 (to J.V.R.), and by a Medical Scientist Training Program grant from the National Institute of General Medical Sciences (T32GM007739 to the Weill Cornell/Rockefeller/Sloan Kettering Tri-Institutional MD-Ph.D. Program). The content is solely the responsibility of the authors and does not necessarily represent the official views of the NIH. The use of the instruments in the Rockefeller University Structural Biology Resource Center was made possible by grant number 1S10RR027037-01 from the National

Center for Research Resources of the NIH. We thank the staff at NSLS-II for their support of remote data collection, in particular Babak Andi for data collection of X0 dataset. This research used resources [AMX and FMX] of the National Synchrotron Light Source II, a U.S. Department of Energy (DOE) Office of Science User Facility operated for the DOE Office of Science by Brookhaven National Laboratory under Contract No. DE-SC0012704. The Center for BioMolecular Structure (CBMS) is primarily supported by the National Institutes of Health, National Institute of General Medical Sciences (NIGMS) through a Center Core P30 Grant (P30GM133893), and by the DOE Office of Biological and Environmental Research (KP1607011). The Rockefeller Structural Biology Resource Center, RRID: SCR_017732, and the use of the Phoenix by Art Robbins in that center was made possible by Grant Number 1S10RR027037-01 from the National Center for Research Resources of the NIH.

## Author contributions

A.G. and K.S.K designed and performed experiments, generated reagents, analysed the data, and wrote the manuscript with input and edits from all co-authors; R.Y. performed in vivo experiments and analysed data; D.A.O, Y.G., J.D., and E.J.S. performed structural analysis and provided intellectual input; P.L. performed analytic ultracentrifugation studies and analyzed the data; J.V.R. designed and directed the study.

## Competing interests

A.G., K.S.K., and J.V.R. have submitted a patent application to the United States Patent Office pertaining to the development and characterization of IgG glycoform-specific nanobodies and methods of use (PCT/US2022/019743). The remaining authors declare no competing interests.
