## [Peer Review File · Nature Communications]

nature portfolio

Peer Review FileEditorial Note: Parts of this Peer Review File have been redacted as indicated to remove third-party material where no permission to publish could be obtained.

REVIEWER COMMENTS

Reviewer #1 (Remarks to the Author):

Gupta and Kao et al present the crystal structure of a novel nanobody, previously generated by the Ravetch group against afucosylated state of human IgG1 Fc, in solution and in a complex with such protein/glycan IgG composition. The study provides detailed structural basis for the unique protein-specific, glycan-restricted binding of this nanobody.

Interestingly, the authors demonstrate that while all CDRs of the nanobody contribute to the binding of the IgG protein, the long flexible CDR3 provides the unique feature of binding specificity toward the buried Fc glycan. This is important study that supports and provides rational on how to design additional nanobodies specific to IgG-glycan complexes and potentially additional glycoproteins. These reagents can be used as diagnostic and therapeutic agents to study and manipulate IgG-Fc glycoforms- a highly desired unmet need in the field. Indeed, this nanobody demonstrated as a therapeutic agent that disrupt the pathogenic role of afucosylated IgG1 complexes in vivo during Dengue virus infection. I have only few minor comments:

1. In figure 2 the authors determine key residues needed for IgG1 Fc binding by referring to reduced binding affinity of X0 alanine mutants. How affinity was determined? This should be specified in the methods.
2. Line 132: Seems there is a missing word after "with", or "with" can just be deleted from the sentence.

Reviewer #2 (Remarks to the Author):

Gupta et al show the structure function relationship between a nanobody recognizing afucosylated IgG. They also show that these nanobodies can prevent antibody-mediated exaggerated inflammation in a model of Dengue Fever. Although the paper is extremely interesting, it currently suffers from a few issues.

Major

- Not showing the specificity issue (fucosylated IgG vs. afucosylated). This is shown in their recently submitted paper but could also be simply be repeated here. In the absence of that, please highly better.
- No figure reports structural coordinates PDB files, or database access. This work can't be fully evaluated without this data.
- please explain the X0's unfavourable binding to afucosylated IgG 3 and 4 (show also for IgG2). Please show subclass dat.

Minor

- Line 63 and general understanding: The core fucose also affects binding to FcγRIIIb.
- Line 65: The elevated binding is also elevated by galactose b,y extra 2 fold (thus for some allotypes of the FcγR, this can be 20 (afucosylated only) -40 fold (with enhanced galactosylation), e.g. Dekkers et al front Immunol 2017
- Lines 68-72) although that association of afucosylated IgG response with pathology shines through here, it may not be clear to all reviewers. Also the fact that at least one paper shows association with protection and afucosylated IgG response for a more latent

virus (HIV, Ackerman/Alter JCI 2014), and another for malaria, would be prudent to add (Larsen Nat Comm 2021)

Reviewer #3 (Remarks to the Author):

This manuscript describes a study on the molecular interaction between IgG1 and a synthetic nanobody X0 specific to afucosylated glycan. Since IgG1 and FcγRIIIa interact by glycan-glycan interface, the interaction between the root GlcNAc and the tip of CDR3 (Gly-Thr) is so impressive. The discussion, i.e., “nanobodies may have a key advantage in recognition of buried glycan structures due to their elongated CDR3 loop”, makes sense to me. Furthermore, the authors performed a mouse infection study to strengthen their conclusion. The text is concisely and clearly written, and the figures are so fine and intuitive. The statistics in Table 1 suggests that the structural quality looks good enough. However, it is necessary to verify whether the crystal structure in the interaction region retains the quality to make such an interpretation, and relevant data should be presented.

Major points:

1. No electron density maps presented. At least, the CDR3 region of X0 and the IgG1 glycan must be shown whether the electron density map was sufficient to model the atomic coordinates.
2. The structures of space group C2 (complex I) and P61 (complex II) are not distinguished. It is not described whether the crystal structure used in the figures is from C2, P61, or which chain in the asymmetric units (there are six in total). The RMSD of 1.5 Å (line 127, C-alpha or all atoms ? Is it between A chains of C2 and P61?) means there are some deviations between the chains, especially for the CDRs.
3. Fig. S3. Please move X0 to the top and indicate residue numbers. I think this figure has essential information to understand the paper, so it should be included in the main figure. Fig. S1 is also a good one to understand the structures of CDRs, and it is worth moving to the main figure.
4. In Fig. S3, mC11 has lower Kd value than X0. It is not mentioned in the text. Did the authors try to crystallize mC11? How about comparing the model structure of mC11 to X0 to find out the reason for the difference in Kd?

Reviewer #1 (Remarks to the Author):

Gupta and Kao et al present the crystal structure of a novel nanobody, previously generated by the Ravetch group against afucosylated state of human IgG1 Fc, in solution and in a complex with such protein/glycan IgG composition. The study provides detailed structural basis for the unique protein-specific, glycan-restricted binding of this nanobody. Interestingly, the authors demonstrate that while all CDRs of the nanobody contribute to the binding of the IgG protein, the long flexible CDR3 provides the unique feature of binding specificity toward the buried Fc glycan. This is important study that supports and provides rational on how to design additional nanobodies specific to IgG-glycan complexes and potentially additional glycoproteins. These reagents can be used as diagnostic and therapeutic agents to study and manipulate IgG-Fc glycoforms- a highly desired unmet need in the field. Indeed, this nanobody demonstrated as a therapeutic agent that disrupt the pathogenic role of afucosylated IgG1 complexes in vivo during Dengue virus infection.

I have only few minor comments:

1. In figure 2 the authors determine key residues needed for IgG1 Fc binding by referring to reduced binding affinity of X0 alanine mutants. How affinity was determined? This should be specified in the methods.

We appreciate the reviewer's comments and we have updated the main text (Line 162) accordingly. In addition we have added a section in the Methods describing our surface plasmon resonance (SPR) protocol.

2. Line 132: Seems there is a missing word after "with", or "with" can just be deleted from the sentence.

We have updated the main text accordingly.

Reviewer #2 (Remarks to the Author):

Gupta et al show the structure function relationship between a nanobody recognizing afucosylated IgG. They also show that these nanobodies can prevent antibody-mediated exaggerated inflammation in a model of Dengue Fever. Although the paper is extremely interesting, it currently suffers from a few issues.

Major

-Not showing the specificity issue (fucosylated IgG vs. afucosylated). This is shown in their recently submitted paper but could also be simply be repeated here. In the absence of that, please highly better.

We appreciate the reviewer's comments. To better clarify this issue of specificity we have provided the affinities of X0 for fucosylated (G2F) and afucosylated (G2) IgG1 (Lines 153-154).

- No figure reports structural coordinates PDB files, or database access. This work can't be fully evaluated without this data.

We agree with the reviewer that database access is important for evaluation of the manuscript. We have released the hold on our PDB entries (8F8V, 8F8W, 8F8X) so that the reviewers may closely examine the data, including structural coordinates and density maps.

- please explain the X0's unfavourable binding to afucosylated IgG 3 and 4 (show also for IgG2). Please show subclass data.

We appreciate the reviewer's interest in IgG subclass specificity across our nanobody clones. Shown below is data from our previous publication (Kao et al, PNAS 2022) demonstrating the affinity of our intermediate affinity clone, B7, for the various afucosylated IgG1-4 subclasses. As shown, our nanobody has affinity for IgG1>IgG2>IgG3>>>IgG4.

In addition, the various IgG subclasses have distinct changes in residues in the BC, C'E, and FG loops (shown below) which contain valuable contact residues for the nanobody-Fc interaction. We have added this alignment data as Fig S3B (shown below) and included a reference to it in the text (line 172).

"[redacted]"

Minor

-Line 63 and general understanding: The core fucose also affects binding to FcγRIIIb.

We have adjusted the text accordingly to include the impact of the core fucose on IgG Fc on binding to FcγRIIIb (Line 63).

-Line 65: The elevated binding is also elevated by galactose by extra 2 fold (thus for some allotypes of the FcγR, this can be 20 (afucosylated only) -40 fold (with enhanced galactosylation), e.g. Dekkers et al front Immunol 2017

We have adjusted the text accordingly (Line 64) and referenced the aforementioned study (Dekkers et al. Front Immunol 2017)

- Lines (68-72) although that association of afucosylated IgG response with pathology shines through here, it may not be clear to all reviewers. Also the fact that at least one paper shows association with protection and afucosylated IgG response for a more latent virus (HIV, Ackerman/Alter JCI 2014), and another for malaria, would be prudent to add (Larsen Nat Comm 2021)

We have adjusted the text accordingly to include scenarios where afucosylated IgG response can be associated with a protective response (Lines 75-76) and referenced the aforementioned study (Larsen et al. Nat Comm 2021) as well as a study from our lab demonstrating the protective effect of afucosylated IgG in a mouse model of viral respiratory illness.

Reviewer #3 (Remarks to the Author):

This manuscript describes a study on the molecular interaction between IgG1 and a synthetic nanobody X0 specific to afucosylated glycan. Since IgG1 and Fc γ RIIIa interact by glycan-glycan interface, the interaction between the root GlcNAc and the tip of CDR3 (Gly-Thr) is so impressive. The discussion, i.e., “nanobodies may have a key advantage in recognition of buried glycan structures due to their elongated CDR3 loop”, makes sense to me. Furthermore, the authors performed a mouse infection study to strengthen their conclusion. The text is concisely and clearly written, and the figures are so fine and intuitive. The statistics in Table 1 suggests that the structural quality looks good enough. However, it is necessary to verify whether the crystal structure in the interaction region retains the quality to make such an interpretation, and relevant data should be presented.

Major points:

1. No electron density maps presented. At least, the CDR3 region of X0 and the IgG1 glycan must be shown whether the electron density map was sufficient to model the atomic coordinates.

We agree with the reviewer that electron density maps are important to evaluate how well our model fits the data. Below, we've included the 2F0-Fc density map contoured at 1.5 sigma of the X0-CDR3 and IgG1-glycan interacting region, which demonstrates that our model fits this interface quite well. We have released the hold on our PDB entries (8F8V, 8F8W, 8F8X) so that the reviewers may closely examine the data, including density maps.

2. The structures of space group C2 (complex I) and P61 (complex II) are not distinguished. It is not described whether the crystal structure used in the figures is from C2, P61, or which chain in the asymmetric units (there are six in total). The RMSD of 1.5 Å (line 127, C-alpha or all atoms ? Is it between A chains of C2 and P61?) means there are some deviations between the chains, especially for the CDRs.

We appreciate the reviewer's comments. We have revised the text to clarify that P61 was used in all the figures presented (Line 128). Chains A, B, C are used for Fig. 2-4. The RMSD of 1.5 Å provided (Line 127) is for all atoms. RMSD between carbon alpha's is 0.963 Å. The main difference between the two structures is that due to the crystal packing there is a domain shift in the IgC1 around the hinge region (residues 342-343). When the N- and C-terminal domains of superimposed separately, the RMSD between CA atoms of (237-342) and (343-443) regions of the two structures are 0.374Å and 0.333Å, respectively. The contact area with the antibody is not affected.

3. Fig. S3. Please move X0 to the top and indicate residue numbers. I think this figure has essential information to understand the paper, so it should be included in the main figure. Fig. S1 is also a good one to understand the structures of CDRs, and it is worth moving to the main figure.

We greatly appreciate the reviewer's comments. While we agree these figures are important to understanding the paper, we do not believe they should be included as main figures. Fig S3 is

intentionally presented in order of increasing affinity, with our parental clone C11 at the top, and our highest affinity clone, mC11, at the bottom. This data was already included in our previous article (Kao et al, PNAS 2022). We have no included residue numbers.

Regarding Fig. S1, we agree that this figure is important in understanding the structures of the CDRs. However, we believe that placing this figure in the main Fig. 1 would interrupt the flow, as there are already bound and unbound structures displayed there. We merely intended to use the figure to display the boundaries of the nanobody CDRs as well as the positions of the key interacting residues, which are more explicitly highlighted in Fig. 2.

4. In Fig. S3, mC11 has lower Kd value than X0. It is not mentioned in the text. Did the authors try to crystallize mC11? How about comparing the model structure of mC11 to X0 to find out the reason for the difference in Kd?

mC11 was described in our previous study (Kao et al, PNAS 2022) and has a Kd of 14.8 nM for afucosylated IgG (G2) and a Kd of 443 nM for fucosylated IgG (G2F). We unfortunately did not try to co-crystallize mC11 with afucosylated IgG1 due to its nonspecific binding profile when generated as a bivalent nanobody-Fc fusion (data not shown). Because of this we do not have a model structure of mC11 to use as a comparison, and simulated structures (i.e. AlphaFold) do not accurately model the CDR loops, as the positions differ drastically from empirically derived structures. Since it is not specifically examined in this manuscript, we are amenable to removing it from Fig. S3A

REVIEWERS' COMMENTS

Reviewer #2 (Remarks to the Author):

I am happy to see very straight to the point answers from the authors, that are very satisfactory.

Reviewer #3 (Remarks to the Author):

The electron density map of the CDR3 is good enough to convince me. However, it is not enough to convince ME with a private letter. The authors must ensure that all readers trust this paper. That is, the figure pasted in the Response letter should be included in the paper as a Supplementary Figure. The answer to my question 2 in the Response letter is very clear, and if it is included in an appropriate place in the main text, the reader will not have any unnecessary doubts and will be able to cite this important paper with confidence. Adding this description will demonstrate that the authors properly understand the protein crystallography. While it is great that the authors have a minimalist spirit and that the paper is so compact, I hope that they accept the above two corrections.

REVIEWERS' COMMENTS

Reviewer #2 (Remarks to the Author):

I am happy to see very straight to the point answers from the authors, that are very satisfactory.

Reviewer #3 (Remarks to the Author):

The electron density map of the CDR3 is good enough to convince me. However, it is not enough to convince ME with a private letter. The authors must ensure that all readers trust this paper. That is, the figure pasted in the Response letter should be included in the paper as a Supplementary Figure. The answer to my question 2 in the Response letter is very clear, and if it is included in an appropriate place in the main text, the reader will not have any unnecessary doubts and will be able to cite this important paper with confidence. Adding this description will demonstrate that the authors properly understand the protein crystallography. While it is great that the authors have a minimalist spirit and that the paper is so compact, I hope that they accept the above two corrections.

We have added the electron density map of CDR3 as Supplementary Figure 4. In addition, we acknowledge the importance of the response included in the point-by-point and have added it into the main text of the manuscript (Line 128-132).